# Gut microbiome signatures in iNPH: Insights from a shotgun metagenomics study

Rahel Park[1,2], Claire Chevalier[1,2], Silas Kieser[1,2], Moira Marizzoni[3], Arthur Paquis[1,2], Stephane Armand[2,4], Max Scheffler[5], Gilles Allali[6], Frederic Assal[2,7], Shahan Momjian[2,7], Giovanni B. Frisoni[1,2]*

1 Memory Center, Department of Rehabilitation and Geriatrics, Geneva University Hospitals, Geneva, Switzerland, 2 Faculty of Medicine, University of Geneva, Geneva, Switzerland, 3 Laboratory of Biological Psychiatry, IRCCS Istituto Centro San Giovanni di Dio Fatebenefratelli, Brescia, Italy, 4 Kinesiology Laboratory, Geneva University Hospitals and University of Geneva, Geneva, Switzerland, 5 Division of Radiology, Geneva University Hospitals, Geneva, Switzerland, 6 Leenaards Memory Center, Department of Clinical Neurosciences, Lausanne University Hospital and University of Lausanne, Lausanne, Switzerland, 7 Department of Clinical Neurosciences, Division of Neurology, Geneva University Hospitals, Geneva, Switzerland

* Giovanni.Frisoni@unige.ch

## Abstract

Idiopathic normal pressure hydrocephalus (iNPH), a leading cause of reversible dementia in older adults, is marked by ventriculomegaly, gait disturbances, cognitive decline, and urinary incontinence. Emerging evidence suggests that gut dysbiosis (microbial imbalance) may influence neuroinflammation and cerebrospinal fluid dynamics, potentially contributing to glymphatic system dysfunction and ventricular enlargement. This study used shotgun metagenomics to analyze the gut microbiome in iNPH patients (n = 18) compared to healthy controls (n = 50), individuals with ventriculomegaly but no iNPH symptoms (n = 50), and Alzheimer's disease patients (n = 50). Microbiome analysis showed an enrichment of species previously linked to various disease states, such as *Enterocloster bolteae* and *Ruminococcus gnavus*, indicating general dysbiosis. In contrast, enrichment of specific taxa, including *Evtepia gabavorous* and *Cuneatibacter sp.*, were specifically associated with iNPH clinical traits, pointing to possible disease-specific microbial markers. Functional analysis showed enrichment of pathways related to carbohydrate and amino acid metabolism, including the S-adenosyl-L-methionine superpathway, implicating inflammatory and immune processes. These findings suggest distinct gut microbiome signatures in iNPH, offering insights into potential gut-brain interactions that may contribute to the disorder's pathophysiology and highlighting possible targets for future therapeutic strategies.

**Data availability statement:** Data are fully available without restriction. The sequences have been deposited at the European Nucleotide Archive, with accession number PRJEB83971. Metagenomic analysis pipeline can be found at https://github.com/SilasK/CDM_pipeline.

**Funding:** This study has been funded by Race Against Dementia, UK (Charity Number: 1165559), under the supervision of the HUG private Foundation. Geneva Memory Center is funded by the following private donors under the supervision of the Private Foundation of Geneva University Hospitals: A.P.R.A. - Association Suisse pour la Recherche sur la Maladie d'Alzheimer, Genève; Fondation Segré, Genève; Fondation Child Care, Genève; Fondation Edmond J. Safra, Genève; Fondation Minkoff, Genève; Fondazione Agusta, Lugano; McCall Macbain Foundation, Canada; Nicole et René Keller, Genève; Fondation AETAS, Genève. The Clinical Research Center, Geneva University Hospitals and Faculty of Medicine, Geneva provides valuable support for regulatory submissions and data management. Funding for this study was awarded to RP. The funders had no role in study design, data collection and analysis, decision to publish, or preparation of the manuscript.

**Competing interests:** The authors have declared that no competing interests exist.

## Introduction

Idiopathic normal pressure hydrocephalus (iNPH) is the leading cause of reversible dementia in older patients. It is caused by altered cerebrospinal fluid (CSF) dynamics, bringing about increased CSF in the lateral ventricles, a distortion of brain tissue, and ensuing neurological symptoms. iNPH is thus characterized by ventriculomegaly and a predominant gait disturbance with the facultative presence of cognitive deficits and urinary incontinence [1]. It has been estimated that approximately 1.3 - 3.7% of the population over the age of 65 is affected by iNPH [2–4]. The onset of iNPH neurological symptoms is preceded by abnormal ventricular enlargement [5,6]. Notably, more than 60% of subjects with asymptomatic ventriculomegaly, characterized by an Evans index >0.3 and a tight high convexity on MRI, developed iNPH over a 16-year longitudinal study [7]. The symptoms of iNPH are similar to other age-related cerebrovascular or neurodegenerative disorders, such as vascular dementia or progressive supranuclear palsy, making the diagnosis complicated and potentially leading to underdiagnosis [3]. When early treated, iNPH symptoms can be reversed. This is achieved by introducing CSF shunts in order to reduce the CSF volume in the brain's ventricles [8].

iNPH is associated with high rates of comorbidity, such as diabetes, obesity and Alzheimer's disease (AD) [9,10]. Brain biopsy studies have shown AD pathology in up to 75% of patients with iNPH, with numbers varying greatly but reaching as high as 89% of AD comorbidity [8]. Furthermore, iNPH and AD patients have in common multiple clinical and pathologic features such as amyloid-β (Aβ) deposits, cerebrovascular inflammation, impaired glymphatic function, and sleep disturbances [11].

Recent evidence suggests that gut microbiomes may play a crucial role in the pathogenesis of various brain disorders via the gut-brain axis [12,13]. Certain gut bacteria can produce neuroactive metabolites, such as short-chain fatty acids (SCFAs) and neurotransmitters, which can affect brain function and contribute to inflammation in the central nervous system. Dysbiosis, or an imbalance in gut microbiota, can lead to increased intestinal wall permeability ("leaky gut"), allowing pro-inflammatory molecules to enter the bloodstream and reach the brain, potentially triggering neuroinflammation. Chronic inflammation is a known factor in neurodegenerative diseases like Alzheimer's and Parkinson's disease. Indeed alterations in gut microbial composition have been linked to Parkinson's disease, with specific bacterial species and their metabolites influencing the gut-brain axis through neuroactive compounds, inflammation, and increased gut permeability [14–16]. Similarly, studies have hinted at a potential association between the gut microbiome and AD [17–19]. A recent study reported genus-level differences in the gut microbiome of iNPH patients compared to healthy controls, based on 16S rRNA amplicon sequencing [20]. However, shotgun metagenomic sequencing offers species-level resolution, greater sensitivity for detecting less abundant taxa, and the ability to predict functional pathways, making it a more powerful and comprehensive approach for microbiome analysis [21].

The aim of this study was to comprehensively examine the microbiome composition, functional pathways, and clinical associations in individuals with iNPH, comparing them to healthy controls and AD patients. Additionally, we recruited participants

with enlarged ventricles, as they might be at a higher risk of developing iNPH. By exploring potential gut-brain interactions in iNPH, this study aims to uncover novel mechanisms underlying this complex condition.

## Materials and methods

### Study participants

The recruitment period for this study extended from December 6, 2017, to April 24, 2024, corresponding to the dates between the first and last sample collected. The study includes participants from two different ongoing cohorts of Geneva Memory Center: the role of gut microbiota in pathophysiology of idiopathic normal pressure hydrocephalus (MIPINO) study, which started in 2022, and the Gut microbiota for Alzheimer's disease (gMAD) study, which started in 2016. These studies were approved by Geneva Ethics Committee (CCER 2022−01059 and 2016−01346).

Patients with suspected iNPH that were referred to the CSF tap-test with an established protocol at the Neurosurgery and Neurology units of Geneva University Hospitals (HUG) and Leenaards Memory Center, were proposed to participate in the study [22]. Briefly, patients undergo a comprehensive evaluation that includes quantitative gait assessment, a neuropsychological test battery, and a comprehensive neurological examination before the CSF tap test (removal of 40 ml of CSF). The following day, patients return for a post-CSF tap test assessment, which includes a repeated quantitative gait assessment and neuropsychological testing focused on cognitive functions commonly affected in iNPH. The diagnosis of possible or probable iNPH is based on criteria outlined in the international consensus guidelines for iNPH and is established before the CSF tap-test takes place [23].

The control group with enlarged ventricles (evHC) was selected from the patients and volunteers of Geneva Memory Center's cohort that already had MRI images available for analysis and were at least 60 years old. The images were processed on the platform provided by NeuGRID2 with FreeSurfer (version 7.1; https://www.neugrid2.eu/). The ventricular size was compared to a normative dataset based on 385 Alzheimer's Disease Neuroimaging Initiative (ADNI) healthy elderly controls. The 80th percentile of the ventricular volume distribution, adjusted for age, was selected as a pragmatic threshold to represent the population with larger ventricular volumes. Subjects with ventricular volumes above this threshold underwent further eligibility testing. Exclusion criteria for this group included: a) Mini Mental State Exam (MMSE) score less than 25, b) substantial brain atrophy (global cortical atrophy scale of 3), c) diagnosis of a neurodegenerative disease or vascular dementia d) diagnosis of a musculoskeletal disorder e) severe or life-threatening disease f) alcohol dependence.

The recruitment coordinator discussed the study details and the participant's role in it with each potential participant. Individuals were given sufficient time to review the Informed Consent Form, ask questions, and give written consent if they decided to take part in the study. When a legal representative was available, the study design, potential risks, and benefits were discussed with them. In the absence of a legal representative, the participant's capacity to understand the study design, risks, and benefits was assessed clinically. Participants with mild or severe dementia were always accompanied by a caregiver, and study participation was discussed jointly with both the participant and the caregiver. The recruitment process is depicted in the flowchart (Fig 1).

AD patients and healthy controls with normal ventricular size (nvHC) were age and sex matched with the recruited participants and selected from the gMAD cohort of Geneva Memory Center. AD patients were selected among amyloid positive cognitively impaired (mild cognitive impairment and dementia) patients. Healthy controls were selected to be either amyloid negative (n = 26) or with no information on amyloid status (n = 24). Amyloid positivity was determined either by visual rating of an amyloid positron emission tomography (PET) scan, by CSF Aβ42 below 880.5 ng/L when measured with Innotest or by an Aβ42/40 ratio below 0.069 when measured with Lumipulse [24,25].

For the evHC group, which included patients and volunteers from the Geneva Memory Center, access to specific clinical variables allowed for a comparison between participants who accepted and those who declined participation (S1 Table in S1 File). The analysis revealed that individuals who accepted were generally younger, had higher levels of education,

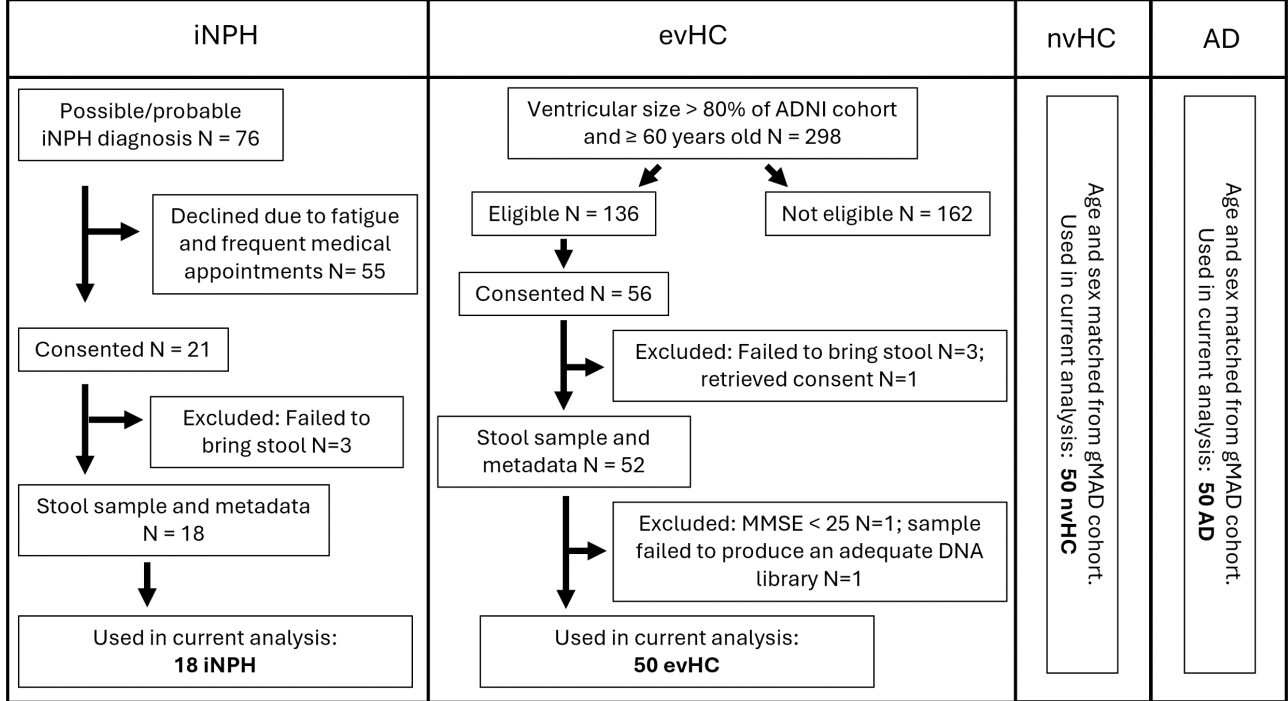

**Fig 1. Recruitment process.** iNPH - idiopathic normal pressure hydrocephalus; evHC - control group with enlarged ventricles; nvHC - controls with normal ventricular size; AD - Alzheimer's disease patients.

and achieved higher MMSE scores. Similar data were not available for eligible iNPH patients, as they were managed by a different unit. Additionally, this type of analysis was not applicable to the AD and nvHC groups, as they were age and sex matched with the recruited groups from a larger cohort.

Subjects who failed to bring in a stool sample were excluded (3 evHC, 3 iNPH). One evHC subject retrieved the consent before sampling, and another had severe cognitive decline since the last visit to Geneva Memory Center with an MMSE score below 25 during the baseline visit, resulting in exclusion. One sample from evHC subject failed to produce an adequate DNA library for sequencing. After the exclusions, the final study groups consisted of 50 evHC, 18 iNPH, 50 nvHC and 50 AD subjects.

## Clinical data, MRI and stool sample collection

**Clinical data.** Participants completed a comprehensive neuropsychological battery. Among the tests performed by a majority of the study participants were MMSE for general cognitive function, Verbal Fluency tests (Fruits) for executive functions, and working memory tests from the WAIS-IV. The evHC and iNPH study groups also underwent gait assessments, involving timed 10-meter walks and the 3-meter Timed Up and Go (TUG) test. Additionally, information on demographics, lifestyle, medical history, anxiety and depression (Hospital Anxiety and Depression Scale; HADS), and urinary symptoms using the Urinary Incontinence Short Form (ICIQ-UI SF) was collected. During the nurse visit the participants underwent physical measurements including heart rate, blood pressure, weight and height for calculating body mass index (BMI).

**MRI imaging.** For the subjects whose MRI scan was conducted more than two years before their enrollment in the study, new baseline MRI scan was performed. MRI exams were conducted at the Division of Radiology, Department of

HUG, using a 3 Tesla Siemens Magnetom Skyra scanner (Siemens Healthineers, Erlangen, Germany) with a 64-channel head coil. 3D T1-weighted images were acquired following guidelines closely aligned with the IMI pharmacog WP5/ European ADNI protocols and previously published procedures [26]. Key parameters for this sequence included a 256 mm field of view, slice thickness ranging from 0.9 to 1 mm, repetition time (TR) between 1819 and 1930 ms, echo time (TE) from 2.19 to 2.4 ms, an 8° flip angle, and no fat suppression.

MRI images were processed with FreeSurfer (version 7.0–recon-all; https://surfer.nmr.mgh.harvard.edu). Hippocampal and ventricular volumes were averaged and normalized as referred to total intracranial volume. The normalized total lateral ventricle volume was determined by first summing the normalized volumes of the left lateral ventricle and the left inferior lateral ventricle, followed by summing the volumes of the corresponding structures on the right side [27]. The mean total lateral ventricle volume was then calculated as the average of the left and right total lateral ventricular volumes.

**Stool sample collection.** The subjects in this study had not had a colonoscopy or received any antibiotic or anti-inflammatory treatment in the month preceding stool donation. Stool samples were collected at home with the use of fecotainer®. The stools were delivered to the Geneva Memory Center within 24 hours of collection, then aliquoted and frozen at either −20°C or −80°C. Freezing temperature was considered as a technical covariate in the analysis.

**Fecal calprotectin quantification.** Fecal calprotectin levels were measured in each stool sample using the Immundiagnostik Calprotectin Enzyme-Linked Immunoabsorbent Assay (ELISA) kit K6927, following standardized protocols.

### Gut microbiota profiling

**DNA extraction, library preparation and sequencing.** DNA was extracted from stools with the MagPurix Bacterial DNA Extraction kit using an automated Zinexts system according to the manufacturer's instructions. Prior to DNA extraction, stools samples were diluted in RLT buffer, mechanically disrupted with bead-beating using Precellys lysing kit soil grinding SK38, 3x40 sec at 6000 rpm, followed by 5 min at 95 °C shaking at 600 rpm. DNA was quantified using a NanoDrop ND-1000 spectrophotometer, and then stored at −20 °C until subsequent analyses. DNA libraries were prepared using the Illumina DNA LP Tagmentation kit following the manufacturer's protocol. Libraries were sequenced on the NovaSeq 6000 DX platform or NovaSeq X Plus aiming for 3.5 GB per sample with 150x2 PE reads (Prebiomics, Italy). Sequencing resulted in an average number of raw reads of 28.53 million, with a standard deviation of 7.25 million.

**Bioinformatic processing.** Firstly, quality trimming was performed on raw sequencing reads using fastp with a base Phred score threshold of 8, a mean quality score of 15, and a minimum read length of 75 bases [28]. Subsequently human reads were removed using Kraken2 (v2.0) with the Human Pangenome Reference Consortium database [29,30]. The taxonomic classification was determined with MetaPhlAn (v4.1) using the CHOCOPhlAnSGB database (vJun23_202307) [31]. Functional gene families and Metacyc pathways were assigned with HUMAnN (v3.9) [32]. The pipeline was implemented using the Snakemake workflow and can be found at https://github.com/SilasK/CDM_pipeline [33].

**Statistical analysis.** In the initial filtering step of the MetaPhlAn taxonomic abundance table, only features classified as bacteria and with at least phylum level assignment were retained. Calculation and visualization of alpha and beta diversities were done using the microViz (v0.12.3) package in R (v4.4.1) [34]. PERMANOVA was performed using dist_calc followed by dist_permanova function from microViz, with 999 permutations. For PERMANOVA, the Bray Curtis distances were calculated from relative abundance data and the Aitchison distances from counts table, both tables filtered for a minimum prevalence of 5%. The differential abundance analysis of the taxa was conducted using the Maaslin2 (v1.18.0) package in R, as well as ANCOM-BC implemented through the QIIME2 (v2024.5) plugin [35–37]. These are widely adopted and validated methods tailored for compositional microbiome data [38]. The Maaslin2 and ANCOM-BC results were corrected for multiple testing using their respective default methods, the Benjamini-Hochberg (BH) procedure and the Holm method. Based on the PERMANOVA results, gender and BMI were used in addition to the study group as fixed effects to account for these confounding variables in the differential abundance analysis. The correlations were

calculated using Spearman method in rcorr function from Hmisc (v5.1-3) package, followed by p-value adjustment with the Benjamini-Hochberg (BH) method, and plotted with heatmap.2 from the gplots (v3.1.3.1) package. The functional output from the HUMAnN pipeline, specifically the normalized (copies per million) MetaCyc pathway abundance table, was analyzed for differential abundance between the study groups using Maaslin2 and ANCOM-BC, with gender and BMI as confounding variables.

## Results

### Cohort characteristics

The 168 participants were divided into four groups: 18 patients diagnosed with iNPH, 50 participants with enlarged brain ventricles (evHC), 50 healthy controls with normal ventricular volumes (nvHC), and 50 AD patients (mild cognitive impairment and dementia). The demographic and clinical characteristics of the study population aligned with expectations (Table 1). Gender distribution was generally balanced but showed a male predominance in the iNPH group. The AD group was slightly less educated compared to others. Neuropsychological evaluations highlighted significant differences across the groups. Notably, while the average MMSE score for the iNPH group remained within the normal range, iNPH is known to significantly impact executive functions [39]. This was evident from the Verbal Fluency test results, where iNPH patients achieved the lowest scores. Working memory impairment was observed in both the iNPH and AD groups; however, statistical significance was reached only in AD patients. Anxiety levels, as measured by the HADS Anxiety Scale, were similar across all groups. In contrast, the prevalence of depression, assessed through the HADS Depression Scale, showed significant variation, with individuals in the evHC and iNPH groups reporting notably higher scores compared to the other groups.

Although none of the self-reported vascular risk factors reached statistical significance, the prevalence of these factors was highest in the iNPH group. This finding aligns with previous reports highlighting the association of vascular risk factors with the disease [10].

Imaging-related metrics revealed significant differences between the groups in terms of hippocampal and ventricular volumes. Notably, hippocampal volume was reduced in the evHC and AD groups, while ventricular volume was increased across all groups compared to healthy controls.

AD is a common comorbidity of iNPH. 38.9% of iNPH patients were amyloid positive, 50% negative, while for 11,1% no data was available. This is slightly lower than the reported 56% of positive AD biomarkers in iNPH patients from the Division of Neurology of HUG between March 2008 and July 2016 [9].

### Brain ventricular volume

Nearly all study participants underwent an MRI scan, enabling us to measure ventricular volumes across study groups and evaluate its utility in differentiating them. The evHC group was defined as individuals with a ventricular volume above the 80th percentile of a healthy elderly population, without iNPH symptoms (see methods). Subjects in the evHC group had volumes above this threshold, while the nvHC group predominantly showed volumes below it, with a few exceptions, in line with a normal distribution (Fig 2A, 2B). Patients with iNPH consistently exhibited ventricular volumes above the 80th percentile, whereas AD patients showed a more evenly distributed pattern around the threshold. Fig 2C shows ventricular volume to be a reliable discriminator among our study groups.

### Microbiome analysis

A total of 168 fecal samples were sequenced to estimate gut microbiota composition. Alpha diversity, which measures the variety of species within a single sample, showed a mild tendency for lower diversity in the evHC, and AD groups compared to nvHC, as indicated by observed species and the Fisher index (S1A Fig). Beta diversity visualization through

**Table 1. Cohort characteristics.**

| Group | nvHC | evHC | iNPH | AD | p-value |
|---|---|---|---|---|---|
| Total | 50 | 50 | 18 | 50 | |
| **Demographics** | | | | | |
| Age | 72.0±5.4 | 72.2±7.7 | 75.5±5.9 | 74.4±5.5 | n.s. |
| Female (%) | 56.0 | 58.0 | 27.8 | 56.0 | n.s. |
| Years of education | 15.3±3.2 | 14.9±4.0 | 14.7±5.4 | 12.3±4.2 | * c |
| **Physical measurements** | | | | | |
| Heart Rate (beats per minute) | 65.5±8.7 | 68.0±9.1 | 67.9±11.9 | 68.8±11.2 | n.s. |
| Diastolic Blood Pressure (mmHg) | 75.7±10.6 | 75.9±9.4 | 79.0±9.9 | 76.5±9.7 | n.s. |
| Systolic Blood Pressure (mmHg) | 130.7±18.1 | 129.4±17.7 | 133.6±17.2 | 139.2±18.2 | * |
| Body mass index | 25.0±3.7 | 26.0±5.8 | 26.4±4.4 | 24.8±4.4 | n.s. |
| **Neuropsychological evaluation** | | | | | |
| Mini Mental State Exam | 28.6±1.1 | 28.3±1.3 | 27.2±2.1 (77.8%) | 23.7±4.3 | **** c |
| Verbal Fluency (Fruits) 2 min | 22.0±4.3 (76%) | 20.0±6.1 | 12.7±4.1 | 15.2±6.0 (74%) | **** b,c |
| Working memory (WAIS-IV) | 24.6±6.4 (76%) | 24.7±5.0 (74%) | 19.7±9.1 (33.3%) | 18.8±7.1 (78%) | *** c |
| **Anxiety and depression** | | | | | |
| HADS anxiety scale | 5.9±3.3 | 7.7±4.4 | 6.5±4.0 | 7.2±4.0 | n.s. |
| HADS depression scale | 3.2±2.8 | 5.4±3.9 | 6.1±3.6 | 3.8±3.0 | ** a,b |
| HADS Total | 9.1±5.6 | 13.1±7.5 | 12.6±7.1 | 11.0±5.5 | * a |
| **Vascular risk factor** | | | | | |
| Atrial Fibrillation (%) | 8.0 | 8.0 | 11.1 | 0.0 | n.s. |
| Cardiovascular disorders (%) | 26.0 (82%) | 22.0 | 22.2 | 26.0 | n.s. |
| Hypercholesterolemia (%) | 28.0 | 32.0 | 50.0 | 42.0 | n.s. |
| Hypertension (%) | 6.0 (70%) | 10.0 (80%) | 22.2 | 10.0 (78%) | n.s. |
| Diabetes (%) | 8.0 | 8.0 | 22.2 | 2.0 | n.s. |
| Smoker (%) | 8.0 | 12.0 | 11.1 | 4.0 | n.s. |
| Alcohol dependence (%) | 8.0 | 8.0 | 11.1 | 0.0 | n.s. |
| **Imaging – MRI** | | | | | |
| Normalized hippocampus (mm$^3$) | 8043±763 | 7214±1059 | 7746±3229 (83.3%) | 6887±1083 | *** a,c |
| Normalized lateral ventricles (mm$^3$) | 14921±5602 | 31840±9087 | 51520±16585 (83.3%) | 23495±8610 | **** a,b,c |
| Callosal angle (°) | NA | 109.7±19.8 | 71.0±23.4 | NA | **** |
| **iNPH characteristics** | | | | | |
| 10m walk time (s) | NA | 10.8±3.6 | 11.7±3.3 | NA | n.s. |
| TUG walk time (s) | NA | 9.4±3.1 | 14.4±3.9 | NA | **** |
| Urinary incontinence score | NA | 3.9±4.5 | 5.5±6.1 | NA | n.s. |

Study groups include: healthy controls with normal ventricle size (nvHC); healthy controls with enlarged ventricles (evHC); idiopathic normal pressure hydrocephalus patients (iNPH) and Alzheimer's disease patients (AD). Data are presented as mean±standard deviation or absolute numbers (percentage), followed by data availability percentage in parenthesis if less than 85% of data was available. Statistical significance, indicated by stars (****$p < 0.0001$, ***$p < 0.001$, **$p < 0.01$, *$p < 0.05$, n.s. not significant), reflects overall group comparisons. These comparisons are performed using one-way ANOVA for numeric variables and chi-square tests for categorical variables. Letters after the stars denote significance relative to nvHC: a for evHC, b for iNPH, and c for AD. Numeric variables were analyzed using t-tests with Bonferroni correction, while categorical variables were analyzed using Fisher's exact test.

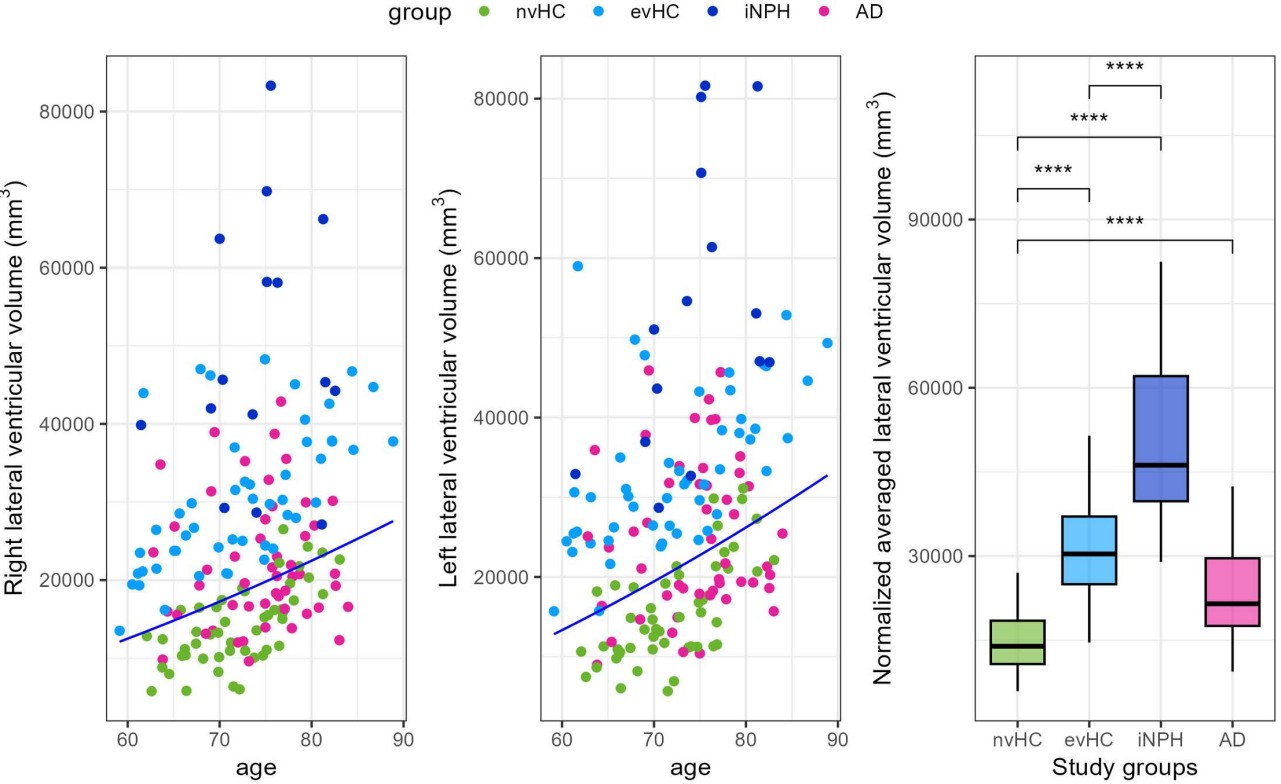

**Fig 2. Brain ventricular volumes by study group.** The left and right lateral ventricular volume of evHC group participants lies above the 80th percentile of the ADNI cohort's normal distribution, represented by a dark blue line **(A,B)**. The study groups differ significantly in brain ventricular volume **(C)**. Statistics: Wilcoxon rank-sum test with **** $p < 0.0001$.

Principal Coordinate Analysis (PCoA), which assesses differences between communities, did not reveal distinct groupings (S1B Fig). Subsequently we performed PERMANOVA to identify whether the following covariates – number of reads, gender, age, BMI, and the study group – could explain the variation in the microbiome composition (S2 Table in S1 File). The number of reads and age did not have a significant effect, whereas the study group, gender (when analyzed using Aitchison distance), and BMI (when analyzed using Bray-Curtis β diversity distance) explained most of the observed variation. The dataset was then subset to include only two study groups at a time to establish which study groups differed the most. PERMANOVA on these subset datasets revealed that the differences were mainly seen between nvHC and evHC groups, as well as between nvHC and iNPH, based on Aitchison distance, which accounts for the compositional nature of microbiome data (Table 2). These differences were not significant when assessed with the Bray-Curtis distance, a measure that emphasizes shared taxa and their relative abundance rather than their compositional relationships.

## Differences in the taxonomic composition of the gut microbiome in iNPH and evHC

To determine the species driving the differences in microbiota composition identified by PERMANOVA, we conducted differential abundance analyses using Maaslin2 and ANCOM-BC. These methods were chosen for their reliability, as they are known to produce consistent results across studies [38]. ANCOM-BC is a more conservative method with lower statistical power, while Maaslin2 offers greater statistical power but may carry a higher risk of false positives.

We identified a total of 43 bacterial species that were either enriched or diminished in the evHC and/or iNPH groups compared to nvHC, based on the combined results of both analytical methods (S3, S4 Table in S1 File). Of these, 14

**Table 2. PERMANOVA analysis of microbiome composition.**

| Group comparisons | β diversity distance | R2 | p-value | Adjusted p |
|---|---|---|---|---|
| nvHC vs evHC | Bray-curtis | 0.0154 | 0.01 | n.s. |
| | Aitchison | 0.0188 | 0.002 | * |
| nvHC vs iNPH | Bray-curtis | 0.0182 | 0.104 | n.s. |
| | Aitchison | 0.0254 | 0.003 | * |
| nvHC vs AD | Bray-curtis | 0.0109 | 0.292 | n.s. |
| | Aitchison | 0.0127 | 0.055 | n.s. |
| evHC vs iNPH | Bray-curtis | 0.012 | 0.887 | n.s. |
| | Aitchison | 0.0144 | 0.512 | n.s. |
| evHC vs AD | Bray-curtis | 0.01 | 0.496 | n.s. |
| | Aitchison | 0.0102 | 0.391 | n.s. |
| AD vs iNPH | Bray-curtis | 0.0139 | 0.634 | n.s. |
| | Aitchison | 0.0161 | 0.233 | n.s. |

Dataset was subsampled for two study groups and evaluated whether the study groups are different in their microbiome, after adjusting for the covariates number of reads, gender, age and BMI. $R^2$ represents the proportion of variance explained by the factors included in the analysis. P-values were adjusted with the Benjamini-Hochberg method.

*adjusted $p < 0.05$; n.s. not significant.

species were identified by both approaches (Fig 3). These consistent differences indicate distinct microbial patterns in the iNPH and evHC groups, in line with the PERMANOVA findings. Notably, ten out of the fourteen taxa exhibited significant shifts in the iNPH group, while the evHC group showed a similar trend, albeit less pronounced. For instance, *Enterocloster bolteae* displayed a log fold change (lfc) of 6.6 in the iNPH group, compared to 3.7 in the evHC group relative to the nvHC group. Importantly, all species identified as differentially abundant in either the evHC or iNPH groups were shifted in the same direction when compared to nvHC. This suggests a gradual transition in microbiome composition from nvHC to evHC and subsequently to iNPH. However, a few taxa, such as *SGB15368*, *Clostridium sp. AF12_28*, and *SGB6334*, showed a more pronounced change in abundance within the evHC group compared to the iNPH group. Species-level genome bins (SGBs), which are reconstructed from metagenomic data, represent microbial taxa that are currently unculturable. Additionally, none of the bacterial species associated with iNPH or evHC reached statistically significant differences in abundance when comparing AD to healthy controls. However, the patterns of bacterial abundance in AD were similar to those in iNPH, with changes occurring in the same direction, albeit without statistical significance (Fig 3). Given the substantial comorbidity between iNPH and AD—more than one-third of iNPH patients are amyloid-positive—these similarities in gut microbiome changes are not unexpected.

## Associations between clinical variables and the identified taxa

In the next step, we correlated the bacterial taxa associated with iNPH and evHC with clinical and imaging variables (Fig 4). Gait measurements and ventricular volume showed strong associations with various bacterial taxa associated with iNPH, and to a lesser extent to taxa associated with enlarged ventricles. Among the taxa that were positively correlated with both brain ventricular volume and gait measurement were *SGB79883*, *Cuneatibacter sp NSJ 177*, *Evtepia gabavorous*, *Anaerotruncus massiliensis* and *Eisenbergiella tayi*. In contrast, urinary incontinence and neuropsychological test scores did not exhibit meaningful correlations with the microbiome. Interestingly, the observed pattern corresponds to the typical progression of iNPH, where brain ventricular enlargement precedes gait disturbance, which is then followed by cognitive deficits and urinary incontinence [7].

Significant correlations with HADS anxiety and depression scores were observed, underscoring the notable prevalence of depression among iNPH patients [40]. Furthermore, we measured calprotectin to investigate the potential role of

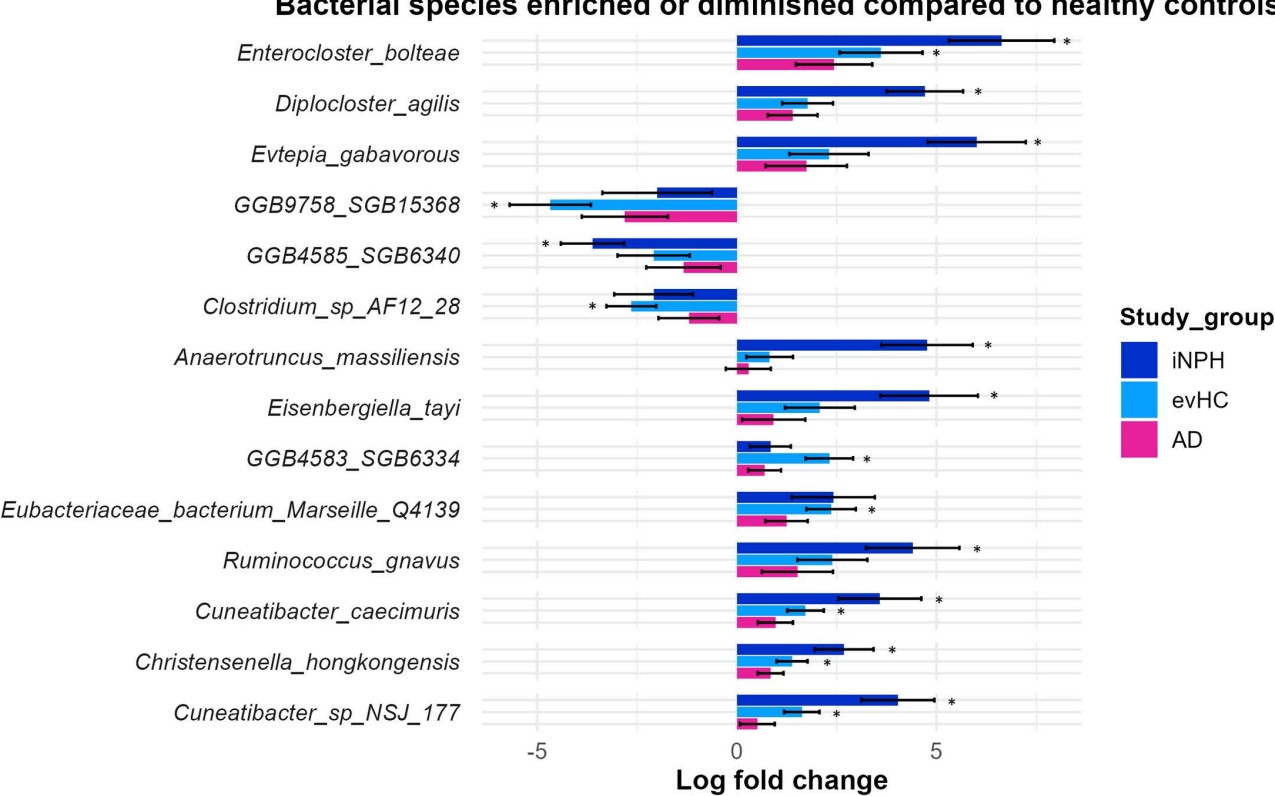

**Fig 3. Barplot showing bacterial species consistently found to be enriched or diminished in study groups compared to nvHC.** Depicted species were detected with both differential abundance methods (Maaslin2 and ANCOMBC, q-value <0.2). Bars denote natural log fold change with standard errors of relative abundances compared to healthy controls (positive values indicate enrichment, negative values indicate depletion), ordered by q-values with ANCOMBC, with the most significant on top. Stars denote taxa with q-values <0.2 with either method in the given study group.

intestinal inflammation in iNPH. Although calprotectin levels did not significantly differ between study groups, there was a clear tendency for a positive correlation with species enriched in the evHC and/or iNPH groups, and for a negative correlation with species that were diminished in these groups. Notably, calprotectin was positively correlated with *Ruminococcus gnavus* and negatively correlated with *Ruminococcus lactaris*.

## Analysis of microbiome functional composition

Shotgun metagenomics allows for the investigation of the functional potential of microbial communities. The differential abundance analysis consistently identified significant pathways, all of which showed increased abundance in the iNPH and evHC groups (Table 3). Notably, pathways related to carbohydrate and amino acid metabolism were substantially enriched, particularly in the iNPH group. Pathways associated with S-adenosyl-L-methionine biosynthesis and L-methionine biosynthesis were highly significant (q < 0.01). In contrast, pathways involved in allantoin degradation were more significantly enriched in the evHC group.

## Discussion

This study presents the first thorough analysis of the gut microbiome in iNPH patients through shotgun metagenomic profiling. We recruited patients with iNPH diagnosis and individuals exhibiting ventriculomegaly, but without clinical symptoms

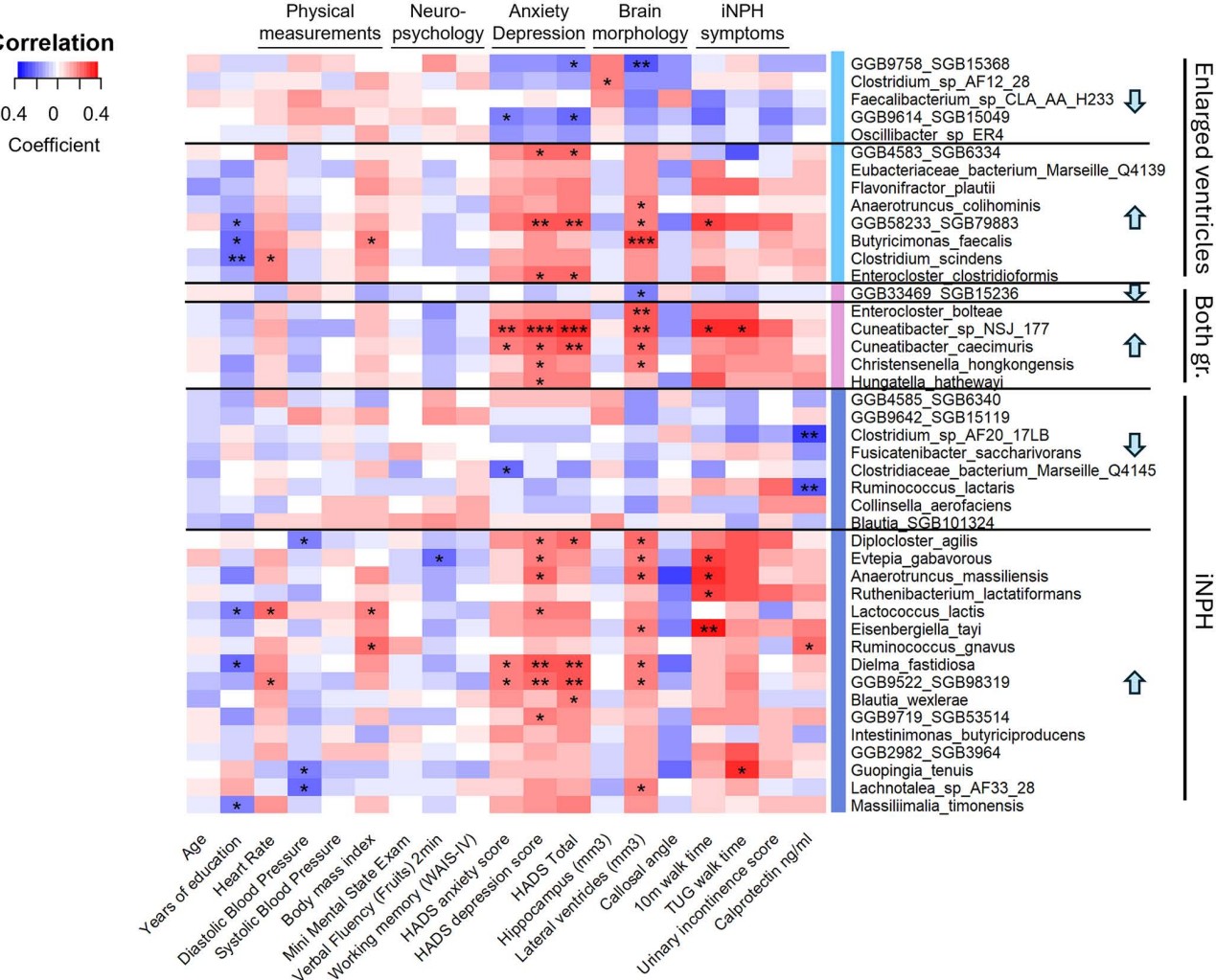

**Fig 4. Spearman correlation heatmap of clinical and imaging variables with bacterial species significantly enriched or diminished in iNPH patients and/or participants with enlarged ventricles.** Differential abundance testing to shortlist bacteria was done with Maaslin2 and ANCOM-BC. Bacterial abundances were CLR transformed prior to Spearman correlation analysis. Taxa marked with light blue, pink or blue bars are differentially abundant in enlarged ventricle, both, or only iNPH study groups, respectively. The arrows indicate whether the taxa is increased or diminished in the given study group. *** adjusted p-value < 0.001, ** adjusted p-value < 0.01, * adjusted p-value < 0.05.

of iNPH. This latter group, referred to as evHC, may represent a population at higher risk of developing iNPH, given that ventriculomegaly often precedes symptom onset [7]. We defined enlarged ventricles as those with ventricular volumes above the 80th percentile of healthy controls, a threshold that appears effective in distinguishing individuals with potential relevance to iNPH progression. Supporting this, the evHC group displayed intermediate characteristics between the nvHC and iNPH groups in several clinical measures, as well as in microbiome composition. For most bacterial species identified in the differential abundance analysis, the log fold changes relative to nvHC were largest in the iNPH group, followed by the evHC group, with changes consistently occurring in the same direction. These findings align with the hypothesis that the evHC group represents a transitional state and highlight the need for longitudinal studies to determine whether these individuals progress to symptomatic iNPH. A longitudinal follow-up study is ongoing and may help to clarify these trajectories.

**Table 3. Metacyc pathways differentially abundant between nvHC and iNPH and/or evHC study groups.**

| Metacyc pathway name | q-value evHC | q-value iNPH |
|---|---|---|
| **Amino Acid Metabolism** | | |
| L-methionine biosynthesis I (HOMOSER-METSYN-PWY) | 0.59143 | ≤ 0.01 |
| Superpathway of S-adenosyl-L-methionine biosynthesis (MET-SAM-PWY) | 1 | ≤ 0.01 |
| Superpathway of L-homoserine and L-methionine biosynthesis (METSYN-PWY) | 1 | ≤ 0.01 |
| Superpathway of L-methionine biosynthesis (transsulfuration) (PWY-5347) | 1 | ≤ 0.01 |
| **Allantoin Degradation** | | |
| Allantoin degradation to glyoxylate II (PWY-5692) | ≤ 0.01 | 0.83552 |
| Superpathway of allantoin degradation in plants (URDEGR-PWY) | ≤ 0.01 | 0.83552 |
| **Carbohydrate Metabolism** | | |
| Hexitol fermentation to lactate, formate, ethanol and acetate (P461-PWY) | 0.32154 | ≤ 0.01 |
| Superpathway of fermentation (PWY4LZ-257) | 0.24944 | ≤ 0.0001 |
| **Xenobiotics Biodegradation and Metabolism** | | |
| Acetylene degradation (anaerobic) (P161-PWY) | 0.20019 | ≤ 0.0001 |
| Superpathway of methylglyoxal degradation (METHGLYUT-PWY) | ≤ 0.05 | ≤ 0.001 |
| **Other Metabolic Pathways** | | |
| Superpathway of N-acetylneuraminate degradation (P441-PWY) | 1 | ≤ 0.0001 |

The pathways listed were identified by ANCOM-BC (q-value < 0.01) and confirmed by Maaslin2, all showing increased abundance compared to nvHC. A full list of combined results can be found in Supplemental S5 Table in S1 File.

In recent years, several large-scale shotgun metagenomic studies have aimed to identify species associated with health and disease using methodologies similar to this study, specifically shotgun metagenomics analyzed with the same pipeline as the one used in this publication (MetaPhlAn) [41–45]. This comparison helped us evaluate how specific our findings are (summarized in S6 Table in S1 File).

Species diminished in the evHC and iNPH groups frequently overlap with taxa associated with healthier profiles in the beforementioned studies. For instance, *Fusicatenibacter saccharivorans* and *Ruminococcus lactaris*, both reduced in iNPH, have been linked to better health outcomes [41,43]. Notably, *R. lactaris* has also been associated with healthy plant-based diet [46]. Similarly, *SGB6340*, depleted in iNPH, correlates strongly with positive dietary habits and cardiometabolic health markers [31]. Another species, *SGB15368*, highly abundant in nvHC but diminished in evHC (S3 Table in S1 File, has been reported to support positive health outcomes [41]. In addition, *SGB15368* shows a strong negative correlation with ventricular volume and HADS total score, highlighting its potential significance and making it a compelling target for further research (Fig 4).

Mechanistic insights from prior research provide context for these observations. For instance, *F. saccharivorans* induces interleukin-10 production, which can mitigate intestinal inflammation [47]. *Oscillibacter* species, another health-linked genus diminished in evHC, metabolize cholesterol, potentially lowering the cardiovascular risk relevant to iNPH pathophysiology [48]. Reduced abundances of these health-associated species could therefore weaken host resilience and contribute to disease mechanisms.

In contrast, several bacteria that are enriched in evHC and/or iNPH have been linked to various disease states (S6 Table in S1 File). These include *Anaerotruncus colihominis*, *Enterocloster bolteae*, *Flavonifractor plautii*, *Hungatella hathewayi* and *Ruminococcus gnavus*, which have been implicated in conditions such as diabetes, an array of gastrointestinal disorders, and fibromyalgia [42]. Some enriched taxa, such as *R. gnavus*, which has been reported in studies of cognitive decline and Parkinson's disease, have been associated with neurocognitive disorders, although findings remain inconsistent [49,50]. Additionally, taxa like *Christensenella hongkongensis*, *Enterocloster clostridioformis* and

*H.hathewayi* are known to occasionally cause bacteremia, suggesting potential roles in systemic infections [51–53]. Among the enriched bacteria, *Evtepia gabavorous* is particularly noteworthy for its capacity to metabolize GABA, a neurotransmitter often deficient in depressed patients [54]. This feature may be relevant to the high prevalence of depression observed in iNPH patients. Some enriched species are less frequently treated in the literature but exhibit significant correlations with iNPH-specific traits. These species include among others *Cuneatibacter* species, *Diplocloster agilis, Anaerotruncus massiliensis,* and *Eisenbergiella tayi*, showing associations with ventricular volume, walking speed and depression, suggesting potential roles in the disease process. This highlights the need for further studies to uncover their functional significance.

Dietary influences are also evident among the enriched bacteria. *E. bolteae* and *R. gnavus* have been linked to fast-food consumption, reflecting unhealthy dietary patterns [55]. Notably, *R.gnavus* was not associated with iNPH related characteristics, but rather with levels of intestinal inflammation marker calprotectin, and BMI (Fig 4). In addition, *R. gnavus* has been described to have mucolytic activity [56]. The mucosal barrier is the host's first line defense against microbial attacks, with mucins being the main structural scaffold. These associations suggest that *R. gnavus* may serve as a marker of general diseased states, rather than being specific to iNPH. In contrast, *R. lactaris*, positively linked to healthy diets, is reduced in iNPH and negatively associated with calprotectin [46]. These opposing trends within the same genus underscore the necessity of species-level resolution in microbiota analyses.

Gut-brain communication may be mediated via multiple pathways, including SCFA production, immune modulation, and maintenance of epithelial and endothelial barrier function along the gut–brain axis [57]. Recent findings have shown that germ-free mice exhibit increased permeability of both the blood–brain barrier (BBB) and the blood–cerebrospinal fluid (CSF) barrier, accompanied by disrupted expression and subcellular localization of tight junction proteins [58,59]. These structural alterations were reversible following recolonization with a normal gut microbiota or supplementation with short-chain fatty acids (SCFAs), highlighting the microbiota's critical role in maintaining barrier function in the central nervous system. Such barrier dysregulation may contribute to impaired CSF homeostasis and immune cell infiltration, mechanisms potentially relevant to the ventricular enlargement and inflammation observed in iNPH [60]. Furthermore, gut dysbiosis has been implicated in chronic neuroinflammation, protein misfolding, and impaired clearance pathways—features common to neurodegenerative conditions including Alzheimer's and Parkinson's disease [61]. However, to the best of our knowledge, the functional dysregulation occurring in iNPH remains completely elusive. Here, we show alterations in microbial signatures in iNPH that may reflect disturbances in gut–brain communication, influencing barrier permeability and neuroimmune signaling.

Functional profiling of the samples revealed significant differences between study groups, primarily in carbohydrate and amino acid metabolism. Several pathways involving superpathway of S-adenosyl-L-methionine (SAM) and methionine metabolism were enriched in the iNPH patients. Disrupted methionine metabolism has been implicated in hepatic, neurological, and cardiovascular dysfunction [62]. In animal models, excessive dietary methionine has been shown to exacerbate neuroinflammation and impair neurogenesis [63]. However, several reports show that lower methionine and SAM levels can lead to neurological abnormalities [64–66]. SAM, a methyl donor for DNA and histone methyltransferases, plays a crucial role in maintaining epigenetic states, and elevated SAM production by gut microbes could influence host gene expression, immune responses, potentially driving disease-associated pathways [67]. Elevated pathways may indicate that the gut microbiota is responding to or contributing to a systemic inflammatory state. Subjects with enlarged ventricles exhibited an increase in allantoin degradation pathways. Allantoin, often produced as a response to oxidative damage, suggests a response to oxidative stress [68,69]. Numerous studies have highlighted the pivotal role of oxidative stress in the development of neurodegenerative conditions, such as Alzheimer's and Parkinson's diseases [70].

Functional predictions were derived from metagenomic profiles without direct metabolomic validation, which constitutes a limitation of the present study. Future work incorporating untargeted metabolomics will be essential to confirm the biological relevance of these pathway enrichments. While this study identified shifts in the microbiome of iNPH patients,

there are several limitations to consider. The primary limitation is the small sample size within the iNPH group, which limits the ability to validate and generalize the findings. We acknowledge that patient recruitment for iNPH is particularly challenging due to the diagnostic complexity of the disease and the physical and psychological burden on patients, which affects research participation. Expanding the cohort would be critical to strengthen the conclusions. Furthermore, the lack of detailed information on dietary intake and medication use restricts our ability to fully interpret the microbiome changes in relation to external factors. Although these data were partially collected, they were not available in complete form or suitable format for reliable analysis across all study groups at this stage. Future studies will prioritize harmonized collection and analysis of these important variables. Another significant limitation is the cross-sectional nature of the study, which precludes us from making causal inferences about the relationship between microbiome alterations and the onset or progression of iNPH.

Longitudinal studies, with follow-up of the defined enlarged ventricle population, are needed to determine whether these microbiome shifts precede the disease or are a consequence of it.

While the identification of disease-associated and health-linked microbial taxa is encouraging, the diagnostic or therapeutic utility of these findings remains to be validated. Future work incorporating longitudinal microbiome tracking, dietary and medication data, and targeted interventions will be key to exploring microbiome-based diagnostics or therapeutics.

## Conclusion

Patients with iNPH exhibit a distinct gut microbiome characterized by an overrepresentation of disease-associated species like *R. gnavus* and *E. bolteae* and a reduction in health-related taxa such as *R. lactaris* and *F. saccharivorans*. Unique associations between certain taxa (*Cuneatibacter sp NSJ 177*, *E. gabavorous*, *A. massiliensis*, *E. tayi*) and iNPH-specific traits, such as increased ventricular volume and impaired mobility, suggest potential disease-specific microbial markers. Functionally, enrichment of methionine and SAM pathways implicates inflammatory and epigenetic processes. These observations highlight a nuanced relationship between the gut microbiome, inflammation, and disease characteristics, with some bacterial species potentially reflecting broader health patterns influenced by diet and other factors. However, validation and further exploration of the identified bacteria are essential to gain actionable insights into their specific roles in iNPH.

## Supporting information

**S1 Fig. Comparison of alpha and beta diversities between the study groups.** Boxplots showing the alpha diversity comparisons between study groups (A). Principal Coordinates Analysis (PCoA) showing the distribution of bacterial communities across study groups, dataset was filtered for 5% minimum prevalence (B).
(TIF)

**S1 File. Supplementary tables.docx Tables S1-S6.**
(DOCX)

**S2 File. metadata_samples.csv.**
(CSV)

## Acknowledgments

We thank all Geneva Memory Center members for their help and suggestions.

## Author contributions

**Conceptualization:** Gilles Allali, Frederic Assal, Shahan Momjian, Giovanni B. Frisoni.

**Data curation:** Rahel Park, Silas Kieser, Max Scheffler.

**Formal analysis:** Rahel Park.

**Funding acquisition:** Giovanni B. Frisoni.

**Investigation:** Rahel Park.

**Methodology:** Rahel Park, Claire Chevalier, Silas Kieser, Arthur Paquis, Stephane Armand, Max Scheffler.

**Project administration:** Rahel Park.

**Software:** Rahel Park, Silas Kieser.

**Supervision:** Claire Chevalier, Moira Marizzoni, Gilles Allali, Frederic Assal, Shahan Momjian, Giovanni B. Frisoni.

**Validation:** Rahel Park, Silas Kieser, Giovanni B. Frisoni.

**Visualization:** Rahel Park.

**Writing – original draft:** Rahel Park.

**Writing – review & editing:** Rahel Park, Claire Chevalier, Silas Kieser, Moira Marizzoni, Arthur Paquis, Stephane Armand, Max Scheffler, Gilles Allali, Frederic Assal, Shahan Momjian, Giovanni B. Frisoni.

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
