## [Decision Letter · Decision Letter 0]

21 Apr 2025

Dear Dr. Frisoni,

Thank you for submitting your manuscript to PLOS ONE. After careful consideration, we feel that it has merit but does not fully meet PLOS ONE’s publication criteria as it currently stands. Therefore, we invite you to submit a revised version of the manuscript that addresses the points raised during the review process.

We look forward to receiving your revised manuscript.

Kind regards,

Jing A. Zhang, MD, PhD

Academic Editor

PLOS ONE

Journal Requirements:

2. Please describe in your methods section how capacity to provide consent was determined for the participants in this study. Please also state whether your ethics committee or IRB approved this consent procedure. If you did not assess capacity to consent please briefly outline why this was not necessary in this case.

3. Thank you for stating the following financial disclosure: [This study has been funded by Race Against Dementia, UK (Charity Number: 1165559)].

Reviewers' comments:

Reviewer's Responses to Questions

**Comments to the Author**

1. Is the manuscript technically sound, and do the data support the conclusions?

Reviewer #1: Yes

Reviewer #2: Partly

2. Has the statistical analysis been performed appropriately and rigorously?

Reviewer #1: Yes

Reviewer #2: Yes

3. Have the authors made all data underlying the findings in their manuscript fully available?

Reviewer #1: Yes

Reviewer #2: Yes

4. Is the manuscript presented in an intelligible fashion and written in standard English?

Reviewer #1: Yes

Reviewer #2: Yes

Reviewer #1: Reviewer Comments

This study provides novel insights into gut microbiome signatures in iNPH using shotgun metagenomics. The findings are promising but would benefit from the instructions below:

(1)The sample size of the iNPH group (n=18) is relatively small, which may limit the statistical power and generalizability of the findings. The authors should explicitly acknowledge this limitation and suggest that future studies with larger cohorts are needed for validation.

(2)Were potential confounding factors (e.g., diet, medication use, comorbidities) adequately controlled for? Since these factors can influence gut microbiota composition, their absence in the analysis may affect the interpretation of results.

(3)The rationale for selecting Maaslin2 and ANCOM-BC for differential abundance analysis should be clarified. Given that microbiome data are compositional, how were false discovery rates (FDR) and multiple testing corrections handled?

(4)The functional analysis suggests enrichment of pathways like S-adenosylmethionine (SAM) biosynthesis, but are these findings supported by independent validation (e.g., metabolomics)? If not, this should be discussed as a limitation.

(5)Several microbial taxa (e.g., Ruminococcus gnavus, Enterocloster bolteae) have been linked to other neurological disorders (e.g., Alzheimer’s, Parkinson’s). How specific are these signatures to iNPH? A comparative discussion with existing literature would strengthen the novelty of the findings.

(6)The correlation between microbial taxa and clinical traits (e.g., ventricular volume, gait disturbance) is intriguing, but can the authors speculate on potential causal mechanisms? For example, could gut dysbiosis contribute to neuroinflammation and CSF dynamics, or vice versa?

(7)The study identifies potential microbial biomarkers for iNPH, but their diagnostic or therapeutic utility remains unclear. Could these findings be translated into clinical applications (e.g., microbiome-based diagnostics or probiotics)?

(8)The gut-brain axis hypothesis is central to this work, but the discussion could be expanded to include possible pathways (e.g., SCFA production, immune modulation, barrier dysfunction).

(9)Figure 3 (barplot of microbial taxa): Consider adding a legend to clarify the direction of change (enriched/depleted) and significance thresholds.

(10)Figure 4 (correlation heatmap): Some labels are unclear—could the axes be better annotated? Additionally, were correlation p-values adjusted for multiple comparisons?

(11)Some sentences are overly complex (e.g., in the Abstract and Discussion). Simplifying phrasing would improve readability.

(12)Ensure consistent formatting of microbial names (e.g., Eviepia gabavorous should be italicized throughout).

Reviewer #2: In this research, shotgun metagenomics was employed to conduct an in - depth analysis of the gut microbiome in 18 patients with iNPH. The results were then contrasted with those of 50 healthy controls, 50 individuals presenting with ventriculomegaly yet without iNPH symptoms, and 50 Alzheimer's disease patients.

The outcomes of this study indicate that there are unique gut microbiome signatures associated with iNPH. These signatures provide valuable insights into the potential interactions between the gut and the brain, which may play a role in the pathophysiology of iNPH. Moreover, they also pinpoint possible targets that could be exploited in future therapeutic strategies for this disorder.

However, several aspects still need to be further improved.

1.The sample size of iNPH patients is relatively small, which may limit the generalizability and reliability of the research findings. A larger sample size would be beneficial to more accurately reflect the real situation of the iNPH population.

2.Additionally, the lack of detailed information on dietary intake and medication use restricts a comprehensive understanding of the influencing factors of gut microbiome changes. It is essential to collect such information to better interpret the relationship between external factors and gut microbiome alterations.

3.Finally, the cross - sectional nature of this study makes it difficult to determine the causal relationship between gut microbiome changes and the onset or progression of iNPH. Longitudinal studies are needed to track the development of individuals over time and clarify these causal links.

**Do you want your identity to be public for this peer review?** For information about this choice, including consent withdrawal, please see our Privacy Policy

Reviewer #1: No

Reviewer #2: No

---

## [Author Response · Author response to Decision Letter 1]

12 Jun 2025

We would like to thank the reviewers and the editor for their thoughtful and constructive feedback. All concerns have been carefully addressed in the accompanying Response to Reviewers file.

---

## [Decision Letter · Decision Letter 1]

30 Jul 2025

Gut Microbiome Signatures in iNPH: Insights from a Shotgun Metagenomics Study

PONE-D-25-08077R1

WAIT FOR NOW

We’re pleased to inform you that your manuscript has been judged scientifically suitable for publication and will be formally accepted for publication once it meets all outstanding technical requirements.

Kind regards,

Jing A. Zhang, MD, PhD

Academic Editor

PLOS ONE

Additional Editor Comments (optional):

Reviewers' comments:

Reviewer's Responses to Questions

**Comments to the Author**

Reviewer #1: All comments have been addressed

Reviewer #2: All comments have been addressed

2. Is the manuscript technically sound, and do the data support the conclusions?

Reviewer #1: Yes

Reviewer #2: Yes

3. Has the statistical analysis been performed appropriately and rigorously?

Reviewer #1: Yes

Reviewer #2: Yes

4. Have the authors made all data underlying the findings in their manuscript fully available?

Reviewer #1: Yes

Reviewer #2: Yes

5. Is the manuscript presented in an intelligible fashion and written in standard English?

Reviewer #1: Yes

Reviewer #2: Yes

Reviewer #1: Gut Microbiome Signatures in iNPH: Insights from a Shotgun Metagenomics Study, after the revision, this article meets the requirements for publication in Plos One

Reviewer #2: The authors have thoroughly addressed the concerns raised in the initial review, and the revised manuscript has significantly improved in terms of scientific rigor and clarity.

The authors acknowledge the small sample size of the iNPH group (n=18) as a key limitation, which aligns with our initial comment. They further explain the challenges in recruiting iNPH patients due to diagnostic complexity and patient burden, providing a reasonable context for this constraint. Importantly, they emphasize that despite this limitation, the findings offer valuable preliminary insights, which is a fair assessment.

The authors openly discuss the lack of comprehensive data on dietary intake and medication use across all groups, noting that these variables were partially collected but not uniformly analyzable. They commit to prioritizing harmonized collection of such data in future studies, which demonstrates awareness of how these factors may influence gut microbiome composition. This transparency strengthens the interpretation of the current results.

The cross-sectional nature of the study, which limits causal conclusions about gut microbiome alterations and iNPH progression, is explicitly acknowledged. The authors’ mention of an ongoing longitudinal follow-up study targeting the enlarged ventricle population is particularly encouraging, as it addresses the need to clarify temporal relationships between microbial shifts and disease onset.

Overall, the revisions adequately address the initial concerns, and the manuscript now provides robust evidence for distinct gut microbiome signatures in iNPH. The findings contribute valuable insights into potential gut-brain interactions in this understudied disorder, warranting publication.

**Do you want your identity to be public for this peer review?** For information about this choice, including consent withdrawal, please see our Privacy Policy

Reviewer #1: **Yes: ** Jing Lu

Reviewer #2: No

---

## [Editor Report · Acceptance letter]

PONE-D-25-08077R1

PLOS ONE

Dear Dr. Frisoni,

I'm pleased to inform you that your manuscript has been deemed suitable for publication in PLOS ONE. Congratulations! Your manuscript is now being handed over to our production team.

Kind regards,

on behalf of

Dr. Jing A. Zhang

Academic Editor

PLOS ONE